# Presynaptic AMPA Receptors in Health and Disease

**DOI:** 10.3390/cells10092260

**Published:** 2021-08-31

**Authors:** Letizia Zanetti, Maria Regoni, Elena Ratti, Flavia Valtorta, Jenny Sassone

**Affiliations:** 1Division of Neuroscience, San Raffaele Scientific Institute, 20132 Milan, Italy; zanetti.letizia@hsr.it (L.Z.); regoni.maria@hsr.it (M.R.); ratti.elena@hsr.it (E.R.); 2Faculty of Medicine and Surgery, Vita-Salute San Raffaele University, 20132 Milan, Italy

**Keywords:** glutamate, GABA, catecholamines, serotonin, acetylcholine

## Abstract

AMPA receptors (AMPARs) are ionotropic glutamate receptors that play a major role in excitatory neurotransmission. AMPARs are located at both presynaptic and postsynaptic plasma membranes. A huge number of studies investigated the role of postsynaptic AMPARs in the normal and abnormal functioning of the mammalian central nervous system (CNS). These studies highlighted that changes in the functional properties or abundance of postsynaptic AMPARs are major mechanisms underlying synaptic plasticity phenomena, providing molecular explanations for the processes of learning and memory. Conversely, the role of AMPARs at presynaptic terminals is as yet poorly clarified. Accruing evidence demonstrates that presynaptic AMPARs can modulate the release of various neurotransmitters. Recent studies also suggest that presynaptic AMPARs may possess double ionotropic-metabotropic features and that they are involved in the local regulation of actin dynamics in both dendritic and axonal compartments. In addition, evidence suggests a key role of presynaptic AMPARs in axonal pathology, in regulation of pain transmission and in the physiology of the auditory system. Thus, it appears that presynaptic AMPARs play an important modulatory role in nerve terminal activity, making them attractive as novel pharmacological targets for a variety of pathological conditions.

## 1. Introduction

Glutamate is the major excitatory neurotransmitter of the central nervous system (CNS). Glutamate can cross the blood–brain barrier through high-affinity transport systems for amino acids. However, most of brain glutamate is locally synthesized through the glutamine pathway [1]. Glutamine is released by glial cells and taken up by neurons where it is transformed into glutamate through the following two different pathways: by the mitochondrial enzyme glutaminase, or by the transamination of 2-oxoglutarate, an intermediate of the tricarboxylic acid cycle. Glutamate is then transported into presynaptic vesicles through a specific transporter (vesicular glutamate transporter, vGLUT). Vesicular glutamate transport is driven by the membrane potential established by the vacuolar H^+^-ATPase (V-ATPase) across the vesicular membrane [2] and results in high intravesicular concentrations of the neurotransmitter. An action potential-induced Ca^2+^ increase in axon terminal induces synaptic vesicle (SV) exocytosis, with an ensuing release of glutamate into the synaptic cleft, where it can interact with glutamate receptors.

Glutamate receptors can be divided into the following two main groups: metabotropic (G-protein coupled receptors) and ionotropic receptors. When glutamate binds a metabotropic glutamate receptor (mGluRs), it activates intracellular signals via interactions with G proteins [3]. In contrast, ionotropic glutamate receptors (iGluRs) are cationic channels.

The following three classes of iGluRs were originally identified based on differential sensitivity to chemical agonists: N-methyl-d-aspartate receptors (NMDARs), α-amino-3-hydroxy-5-methyl-4-isoxazolepropionic acid receptors (AMPARs) and kainate receptors (KARs). Glutamate binding promotes a conformational change in their quaternary structure, allowing ions such as Na+, K+ and Ca^2+^ to flow through the channel pore.

AMPARs are multimeric assemblies of GluA1-4 subunits (formerly named GluR1-4). GluA1-4 proteins are encoded by *GRIA1-4* (Glutamate Receptor Ionotropic AMPA types 1–4) genes localized in the human genome on chromosomes 5q33.2, 4q32.1, Xq25 and 11q22.3, respectively [4]. Subunits have a similar size (approximately 900 amino acids each) and share 68–73% of their amino acid sequence identity.

Each AMPAR subunit contains an extracellular N-terminal domain (NTD), a ligand binding domain (LBD), a transmembrane domain (TMD) that forms the ion channel and a C-terminal cytoplasmic domain (CTD). The TMD contains three membrane-spanning helices (M1, M3 and M4) and a membrane re-entrant loop (M2). The NTD folds into a bilobate structure and encompasses about 50% of the length of the entire subunit; it is involved in receptor assembly and plays a crucial role in AMPAR anchoring at synapses (Figure 1). While NTD-deleted AMPARs are fully functional, they exhibit slightly altered desensitization kinetics, in keeping with the notion that this domain can influence the adjacent LBD [5].

The LBD is the site where the agonist is docked. It exhibits a bilobate structure and captures the ligand in an interlobe cleft, as its D1 and D2 domains close similarly to a clamshell structure. The D1 and D2 domains are created by the proximity of two segments of amino acids termed S1 (the first 150 amino acids of M1) and S2 (the extracellular portion between M3 and M4) domains. Since the LBDs of the adjacent subunits dimerize back-to-back via their upper (D1) lobes, the closure of the clamshell around glutamate causes the separation of the lower (D2) lobes, which transmit the mechanical force for the opening of the channel gate, followed by rapid desensitization [6]. M2 cannot be considered as a real transmembrane domain, as it does not completely span the cytoplasmic membrane, and shrinks at the base of the pore with a diameter of 0.7–0.8 nm [7].

The intracellular CTDs of AMPAR subunits have variable length: GluA1 and GluA4 have longer cytoplasmic tails, whereas GluA2 and GluA3 have shorter cytoplasmic tails [8]. These domains are involved in auxiliary protein interactions, which can partially differ depending on their length and composition. Hence, they influence the electrophysiological properties of the channel in a partially distinct manner [6].

Alternative splicing generates two splice variants for each subunit, named “flip” and “flop”. These subunits are generated from adjacent exons, each encoding a module of 38 amino acids in a conserved receptor domain located between the third and fourth transmembrane segments. Interestingly, they are believed to control the rate of the channel closing process [9]. Figure 1 illustrates the AMPAR structure (Figure 1).

AMPARs exist as homo- or heterotetramers of various combinations of the four subunits, thus giving rise to large receptor diversity. The subunits are combined with various stoichiometry [10]; each subunit contributes to conferring the channel various kinetics, ion selectivity and trafficking properties, all together defining the receptor functions.

The GluA2 subunit exists in distinct edited and unedited forms, GluA2(R) and GluA2(Q), which differ in a single amino acid at position 607 in the transmembrane segment M2 (Q/R site). This alteration of charge, which occurs in the channel pore, blocks the passage of Ca^2+^ ions. Hence, only AMPARs either lacking GluA2 or containing unedited GluA2(Q) are Ca^2+^ permeable [11]. Since in the brain the vast majority (>99%) of GluA2 mRNA exists in the edited form GluA2(R), the absence of the GluA2 subunit can endow the channel with permeability to Ca^2+^, in addition to Na+ and K+, with consequences on plasticity phenomena and on the vulnerability of neurons to excitotoxicity.

The receptors composed of GluA1 assembled with GluA2, and GluA2 assembled with GluA3 are the most abundant forms in the adult hippocampus [12]. In the adult nucleus accumbens, dorsal striatum and prefrontal cortex, a predominant role for GluA1/2 receptors and a smaller role for GluA2/3 receptors was described [13].

Distinct AMPARs are responsible for long-term potentiation (LTP) and receptor turnover: in particular, at the postsynapse of hippocampal neurons, the various AMPARs display distinct synaptic delivery mechanisms. The receptors composed of GluA2 and GluA3 continuously replace synaptic GluA2/3 receptors in a manner that maintains baseline transmission, whereas GluA1/2 receptors are added to synapses during plasticity [8,14].

The aim of this review is to collect the scientific evidence about presynaptic AMPARs. The characteristics of postsynaptic AMPAR and the large number of associated proteins that regulate the trafficking/localization/channel properties of this receptor have been extensively described in several recent reviews [6,11,15].

## 2. AMPAR Localization at Presynapse

AMPARs were revealed at the presynaptic terminals of several neuron types. Fabian-Fine and coauthors investigated GluA2/3 labelling at presynaptic sites in rat hippocampal slices using transmission electron microscopy (TEM). They found that GluA2/3-immunoreactive gold-particles often colocalized at single vesicular structures within the presynaptic element. In subcellular fractions of hippocampus the GluA2/3 immunoreactivity comigrated with the SV marker synaptophysin. These data corroborated the presence of AMPARs in presynaptic vesicular structures [16]. The purification of small SVs from rat forebrain homogenates through chromatography confirmed that AMPAR subunits (GluA1 and GluA2) co-purified with SVs [17]. Another study performed in rat cortex and hippocampi investigated the localization and modulation of AMPAR subunits GluA1-2-3 in the presynaptic fractions and in the postsynaptic density in basal condition and upon stimulation with the AMPA/NMDA agonists. AMPA stimulation resulted in a marked decrease in postsynaptic GluA2 and GluA3 subunits, conversely, GluA2 and GluA3 levels increased in the presynaptic fraction. No significant changes in any of the compartments were detected for the GluA1 subunit. NMDA treatment decreased postsynaptic GluA1 and GluA2 but increased the presynaptic levels of these subunits. NMDA treatment did not evoke changes in GluA3 localization. The authors concluded that presynaptic and postsynaptic GluA1 and GluA2 subunits are regulated in opposite directions by AMPA or NMDA stimulation [18].

Presynaptic AMPARs were also investigated in cerebellar molecular layer interneurons (MLIs). Rossi and coauthors demonstrated using single-cell RT-PCR experiments that GluA1–4 subunits are expressed in the axons of MLIs. These presynaptic AMPARs elicited cytosolic [Ca^2+^] transients in axonal varicosities following the glutamate spillover induced by the stimulation of parallel fibers. According to this study, presynaptic AMPARs support a powerful potentiation of GABAergic synaptic transmission in cerebellar interneurons [19].

The presynaptic localization of AMPARs was also investigated in dorsal root ganglion (DRG) neurons. Lu et al. reported a conspicuous and selective expression of presynaptic AMPAR subunits in terminals of different types of primary afferents in the superficial laminas of the dorsal horn of the spinal cord. The staining for AMPAR subunits was superimposed with staining for synaptophysin, thus suggesting a presynaptic localization of AMPARs in the dorsal horn. Puncta immunopositive for synaptophysin and GluA2/3 were located predominantly in laminas III and IV, whereas puncta immunopositive for synaptophysin and GluA4 or GluA2/4 were located predominantly in laminas I and II. These results suggest a role of AMPARs in the modulation of neurotransmitters’ release by terminals of both myelinated and unmyelinated afferents of DRG neurons [20]. Presynaptic AMPARs were also identified in myelinated rat dorsal column fibers. The immunohistochemical characterization of GluA4 in dorsal columns revealed frequent punctate staining at the surfaces of neurofilament-positive axon cylinders and immuno-electron microscopy confirmed that many GluA4-positive clusters were localized at the internodal axolemma [21].

AMPARs were found also on axon terminals of corticostriatal and thalamostriatal afferents. Preliminary results from Bernard and coauthors initially suggested a presynaptic localization of AMPARs in the neostriatum through immunoperoxidase electron microscopy [22]. Another study performed freeze-postembedding immunogold labeling on neostriatum rat sections in order to obtain double labelling for vGLUTs and glutamate receptors; these experiments highlighted that vGLUT1- and vGLUT2-positive axon terminals in the neostriatum were also labeled for GluA1–4 subunits and suggest that the glutamate released from the axon terminals of corticostriatal and thalamostriatal afferents controls the activity of the terminals through presynaptic AMPA autoreceptors [23].

Functional AMPARs are also expressed in the calyceal nerve terminal (calyx of Held) of rodents. Here, their activation inhibits voltage-gated Ca^2+^ currents through coupling with heterotrimeric G proteins [24]. At the calyx of Held, presynaptic AMPARs also showed an ionotropic nature, depolarizing the nerve terminal to levels that induce increases in neurotransmitter release; however, according to the authors, this facilitatory effect is masked by the stronger inhibitory effect mediated by G proteins [24]. These results support a physiological role of presynaptic AMPARs at the calyx of Held in vivo.

Interestingly, the presynaptic localization of AMPAR subunits was reported to be regionally and developmentally regulated. By TEM, GluA1 was found in both the dendritic and axonal terminal components of developing synapses, specifically associated with the tubulovesicular cisternal compartment that characterizes growth cones. The GluA1 presence at presynaptic sites dissipated with synaptic maturation, becoming confined to the somatodendritic compartment as maturation progressed [25]. Another study investigated the presence of AMPAR subunits and the possible dynamic control of their surface exposure at the presynaptic membrane in the axonal growth cones of hippocampal neurons grown in vitro (5–7 days in vitro). The percentage of GluA2/3-positive growth cones was 77 ± 14%. The percentage of GluA1-positive growth cones was 42 ± 7.3% [17].

These data were further supported by the evidence provided by Wyszynski et al. [26], which described the interaction of AMPAR with Liprin-α-GRIP, a protein complex whose function is crucial in presynaptic development and differentiation, acting as a structural component of the active zone or as a molecular anchor recruiting other molecules to the active zone [27].

The mechanisms governing the trafficking of presynaptic AMPARs were also investigated. Studies on hippocampal noradrenergic axon terminals showed that AMPAR activation evokes noradrenaline (NA) release and that this effect is prevented by blocking the interaction between the subunit GluA2 and the protein interacting with C kinase 1 (PICK1); coherently, the authors suggested that PICK1 is involved in the regulation and trafficking of presynaptic GluA2 subunits [28]. A subsequent study investigated the interaction between PICK1 and GluA2: Haglerød and coauthors found that GluA2 and PICK1 are detectable in both synaptosome and SV fractions from rat brain fractions. In the same fractions, GluA2 coimmunoprecipitated with PICK1; lastly, double immunogold labeling experiments showed that PICK1 and GluA2 colocalized at the presynaptic plasma membrane and SVs along the active zone [29].

PICK1 and GluA2 was found in association with the presynaptic plasma membrane and SVs in hippocampal excitatory synapses [29]. These findings were confirmed and deepened by another study from the same authors. The immunogold labeling of sections of the hippocampal CA1 region with an anti-GluA2 antibody confirmed that the GluA2 subunit and PICK1 colocalized at the active zone. Moreover, the overexpression of GFP-PICK1 by the viral infection of organotypic hippocampal cultures significantly reduced the GluA2/3 immunogold labeling within the active zone and in the presynaptic cytoplasm [30]. These data suggest that PICK1 acts as regulator of presynaptic AMPAR trafficking.

In order to estimate the abundance of the presynaptic AMPAR as compared to the classic postsynaptic AMPAR, Feligioni and coauthors investigated the ultrastructural localization of the AMPAR in synaptosomes prepared from rat cortices and hippocampi. All the AMPAR subunits were enriched in the postsynaptic fraction but were also detectable in the presynaptic and non-synaptic synaptosome fractions (NSSP), with considerable subunit variability in the amount of immunoreactivity among the various fractions. GluA1 was relatively abundant in the NSSP fraction (~20%) with comparatively similar proportions (35–45%) present in the pre- and postsynaptic fractions. Only low levels of GluA2 were detected in the NSSP fraction, whereas this subunit represented ~60 and ~35% of the subunits in the post- and presynaptic fractions, respectively. GluA3 was mainly localized in the postsynaptic fraction and was not detected in the NSSP fraction [18]. These data suggest that the levels of the presynaptic AMPAR are roughly comparable to the levels of the postsynaptic AMPAR in the cortex and hippocampus; however, in other brain regions, the abundance of the presynaptic AMPAR as compared to the classic postsynaptic AMPAR remains to be clarified.

Altogether, these studies provide strong evidence for a presynaptic localization of the AMPAR in various neuronal types, supporting the hypothesis that presynaptic AMPARs play a key role in the early stages of neuronal development, and provide clues about the molecular mechanisms regulating presynaptic AMPAR trafficking. In addition, the presence of AMPARs, both at the active zone and in SVs, suggests the existence of a reserve pool of AMPARs that can be recruited to the presynaptic membrane depending on physiological needs.

## 3. Physiological Functions of Presynaptic AMPAR

In the absence of AMPA agonists/antagonists selective for the presynaptic receptors, to define the role of AMPARs at the presynapse is a challenging task. Moreover, it is difficult to discriminate whether changes in the evoked or spontaneous synaptic transmission are due to the activation of presynaptic AMPARs or to indirect effects deriving from glutamate. However, several lines of evidence suggest that AMPAR localization at the presynapse may modulate neurotransmitter release from the axon terminals of various fiber types, raising the possibility of an autocrine or paracrine feedback to both excitatory and inhibitory terminals.

### 3.1. AMPAR as Regulator of Neurotransmitter Release

#### 3.1.1. Glutamate

A wide range of physiological and pathological processes including synaptic plasticity, epileptogenesis and ischaemic brain damage rely on the modulation of glutamate release; presynaptic AMPAR may play a crucial role in these mechanisms. Patel et al. were the first to suggest a role for AMPAR in a positive feedback control of glutamate release at the synapses. Rat forebrain slices were preloaded with [^3^H]-D-aspartate, a non-metabolized L-glutamate analogue, then the Ca^2+^-dependent release of [^3^H]-D-aspartate was measured in basal conditions and after the administration of either L-glutamate or racemic (R,S)-AMPA (activity resides solely in (S)-isomer). L-glutamate and racemic (R,S)-AMPA dose-dependently enhanced [^3^H]-D-aspartate release. The response to receptor stimulation was potentiated by cyclothiazide, a lectin that blocks AMPAR desensitization, and is inhibited by competitive and non-competitive AMPAR antagonists [31]. In a further study, the same group used intracerebral microdialysis to test whether presynaptic AMPAR plays a role in the control of glutamate release in vivo. AMPA application increased the Ca^2+^-dependent efflux of both [^3^H]-glutamate and [^14^C]-GABA released from the rat neostriatum in a dose-dependent manner. However, the AMPAR desensitization inhibitors, cyclothiazide or aniracetam, did not potentiate the effect. The authors interpreted these data as suggestive of the presence of presynaptic AMPARs of a novel cyclothiazide- and aniracetaminin-sensitive subtype on presynaptic nerve terminals in the rat striatum [32]. Another study investigated the role of AMPARs in primary afferent fibers (PAFs) of DRG neurons. By recording the receptor-mediated depolarization of the central terminals, the authors showed that AMPARs have a strong inhibitory action on the amount of glutamate released from primary afferent terminals in the superficial layers of the dorsal horn. The authors hypothesized that the mechanism underlying this effect is primary afferent depolarization (PAD). PAD is a mechanism that can lead to action potential shunting by creating a local area with elevated input conductance, which decreases the magnitude of depolarization during the action potential and slows the action potential propagation. Furthermore, AMPAR-mediated PAD may inactivate voltage-gated sodium channels, making the action potential less likely to occur. These results challenged the traditional view stating that only GABA receptors mediate PAD, indicating that PAD can also be mediated by glutamate acting on presynaptically localized AMPARs [33].

Following reports showing that AMPARs can be upregulated by protein kinase A (PKA) via phosphorylation [34], as well as influenced by intracellular Ca^2+^ and calmodulin (CaM) levels [35], Dohovic et al. hypothesized that the modulation of presynaptic iGluRs by the cyclic adenosine monophosphate (cAMP)-dependent phosphorylation cascade is important for the regulation of glutamate release. Using superfused mouse striatal slices, they administered D-[^3^H]aspartate with either isoproterenol or propranolol (β-adrenoceptor agonist and antagonist, respectively). β-adrenergic receptor-mediated adenylyl cyclase activation and cAMP accumulation were found to potentiate AMPA-evoked glutamate release from striatal glutamatergic potentials [36].

Overall, experimental evidence strongly suggests that presynaptic AMPARs can regulate glutamate release and may play an important role in a range of physiological processes, including synaptic plasticity and the control of membrane potential.

#### 3.1.2. GABA

One of the best studied functions of the presynaptic AMPAR is the regulation of GABA release in the cerebellum. Bureau and Mulle investigated GABA release from cerebellar interneurons and found that AMPAR activation by low concentrations of domoate potentiated GABAergic synaptic activity in stellate cells. This effect can be explained by a presynaptic modulation of GABAergic transmission due to the axonal presynaptic localization of AMPAR. The domoate effect was observed at immature stages during the development of the stellate cells, but not at more the mature stages (after postnatal day (P) 21) [37].

Fiszman and coauthors provided further functional evidence that AMPARs are present in developing GABAergic terminals of cerebellar interneurons and that their activation affects the size of GABAergic terminals and spontaneous GABA release. By using primary cultures of cerebellar neurons derived from mice transgenic for GFP-GAD65, that allows the direct identification of GABA neurons, they found that either AMPA or kainate (which acts as an agonist on both KARs and AMPARs) induced a pronounced change in the size of presynaptic GABAergic terminals of interneurons. The effects of both the excitatory amino acid receptor agonists were inhibited by the selective AMPAR antagonist GYKI52466. Whole-cell recordings confirmed the functional relevance of these morphological changes. Indeed, kainate increased the spontaneous synaptic release from GABAergic interneurons in cerebellar cultures [38].

Another study explored the interactions between cerebellar excitatory and inhibitory synapses in rat cerebellar slices. Satake et al. found that the glutamate released from the climbing fiber (CF) is responsible for both the direct excitation of Purkinje cells (PCs) and the inhibition of GABAergic transmission from interneurons converging on the same PCs. The authors suggested that glutamate released into the CF-PC synapse can spill over and act on AMPARs located in neighboring presynaptic terminals of interneuron basket cells (BCs), thus resulting in the inhibition of GABAergic activity [39] (Figure 2). The same authors investigated the underlying molecular mechanism and found that glutamate can mediate the presynaptic suppression of inhibitory GABAergic transmission through the AMPAR-mediated activation of GTP-binding proteins (G_i/o_ proteins). In particular, by recording inhibitory postsynaptic currents (IPSCs) from PCs and following intracellular Ca^2+^ fluctuations at BC terminals, they found that AMPAR activation induced dissociation of βγ subunits from G-proteins and inhibited the activity of P/Q-type Ca^2+^ channels in the nerve terminals of cerebellar interneurons [40]. These findings were confirmed by Rusakov et al., who provided the first qualitative description of presynaptic Ca^2+^ kinetics and its modulation by AMPAR activation occurring via a glutamate spillover-mediated mechanism at GABAergic synapses [41]. Taken together, these studies proposed a new double function of glutamate released from the CF: on one side, it activates “ionotropic” AMPARs eliciting postsynaptic excitation at CF–PC synapses; on the other side, it activates “metabotropic” AMPARs mediating presynaptic inhibition at BC–PC inhibitory synapses [40] (Figure 2). Since the long-term plasticity of cerebellar synaptic transmission is seen as the primary motor learning mechanism, we can speculate that presynaptic AMPAR activation is one of the molecular events contributing to this mechanism.

Presynaptic AMPAR is also involved in the control of spinal nociception. Engelman et al. provided morphological and functional evidence for presynaptic AMPARs in GABAergic interneurons in the spinal cord dorsal horn. GABAergic interneurons provide local inhibition within the spinal cord dorsal horn and strongly influence pain and temperature signaling. Recording evoked and miniature inhibitory postsynaptic currents (eIPSCs and mIPSPs, respectively) in the superficial dorsal horn of the rat spinal cord at the second postnatal week, these authors showed that AMPAR activation increases the spontaneous release of GABA onto both lamina II and Neurokinin-1 (NK1) receptor-expressing (NK1R+) lamina I neurons. The GABA release was sensitive to the extracellular concentration of Ca^2+^, suggesting a possible Ca^2+^ entry through AMPARs. AMPA-induced increases in the mIPSP frequency were also observed in dorsal horn slices prepared from P27–P30 rat spinal cord. Since inhibition in the dorsal horn is important for controlling nociceptive signaling, the authors suggested that presynaptic AMPARs can modulate pain signaling in the spinal cord dorsal horn [42].

#### 3.1.3. DA, NA, 5HT and ACh

Experimental evidence suggests that presynaptic AMPARs can regulate the release of a wide range of neurotransmitters in addition to GABA and glutamate, including NA [43], dopamine (DA) [44,45], serotonin (5HT) [46] and acetylcholine (ACh) [47,48,49,50]. These phenomena can occur in various brain areas, such as the corpus striatum, hippocampus, nucleus accumbens, prefrontal cortex and olfactory bulb [31]. A potential caveat to these studies is that they did not distinguish the effects deriving from the activation of AMPARs from those deriving from the activation of KARs, because they employed agonists and antagonists that did not allow such distinction. With this caveat in mind, we review here the literature supporting a presynaptic facilitation of DA, NA, 5HT and ACh release induced by AMPARs/KARs.

Using purified synaptosomes from rat striatum superfused with [^3^H]-tyrosine in order to estimate the release of newly synthesized [^3^H]DA, Desce et al. showed an increase in [^3^H]DA release upon stimulation with kainate and quisqualate (agonist of AMPARs and KARs). This facilitation of DA release was potentiated by concanavalin A [44,51], a lectin able to strongly potentiate the KAR current and to weakly potentiate the AMPAR current [52]. This result suggested that presynaptic AMPARs/KARs are involved in the regulation of DA release. A further study explored the modulation of the release of [^3^H]DA and [^3^H]NA induced by presynaptic glutamate receptors in hippocampal synaptosomes. Using a superfusion system and a fluorometric assay with Indo-1 as a probe for [Ca^2+^], the authors found that presynaptic glutamate receptor activation induced a dose-dependent release of [^3^H]DA and [^3^H]NA. Furthermore, the application of N- or P-type (but not L-type) voltage-sensitive Ca^2+^ channel (VSCC) blockers strongly inhibited the release of catecholamines in this experimental setting, suggesting that the membrane depolarization induced by AMPAR activation is coupled with an increase in cytosolic [Ca^2+^] [43]. In yet another study, rat hippocampus synaptosomes were found to release preloaded [^3^H]NA in a Ca^2+^-dependent manner when challenged with increasing AMPA concentrations [53].

Fink et al. investigated the role of presynaptic glutamate receptor in 5HT release in rat brain cortex slices, showing that the activation of NMDA and non-NMDA receptors elicited a release of 5HT. Slices were preincubated with [^3^H]5HT and then superfused in the presence of 6-nitroquipazine (an inhibitor of the 5HT transporter, in order to block the reuptake of [^3^H]5HT). When superfused with a solution containing Mg^2+^, kainic acid (KA) or AMPA, a concentration-dependent overflow of tritium was recorded. The kainate-evoked tritium overflow was inhibited by the AMPAR/KAR antagonist 6-cyano-7-nitroquinoxaline-2,3-dione (CNQX), but not affected by the NMDA receptor antagonist CGP37849. Interestingly, the amount of 5HT released in response to KA and AMPA was lower as compared to the amount evoked by NMDA. The authors argued that this effect is probably due to rapid desensitization, a typical feature of ionotropic AMPAR [54]. Another study performed similar experiments in hippocampal synaptosomes. The authors showed that superfusion with AMPA provoked a Ca^2+^-dependent and 6,7-dinitroquinoxaline-2,3-dione (DNQX)-sensitive release of [^3^H]5HT, thus confirming the AMPAR/KAR role in mediating the release of 5HT [55].

In a preliminary study, Lupp and coworkers incubated slices of rabbit caudate nucleus with [^3^H]ACh and found that stimulation with NMDA, AMPA, L-glutamate or KA caused an increase in [^3^H]ACh efflux. This release was Ca^2+^-dependent and tetrodotoxin-sensitive [47]. Another study in rat striatal slices preincubated with [^3^H]ACh showed that 4-aminopyridine (4-AP, a K+ channel blocker that stimulates the release of glutamate) increased ACh release [48]. Morari and coauthors investigated the role of the glutamatergic modulation of striatal cholinergic transmission in rat striatal slices and synaptosomes. They compared the effects of NMDAR and AMPAR on endogenous ACh release and found that both NMDA and non-NMDA ionotropic receptors, localized on the somatodendritic complex and on the axonal terminals of striatal cholinergic interneurons, facilitated striatal ACh release. However, they also showed that the stimulation evoked by AMPA was followed by a prolonged inhibition of ACh release [49]. Interestingly, another paper provides a possible explanation about this latter evidence. The authors used a transversal microdialysis probe implanted into the striatum of freely moving rats. Due to the restricted diffusion of the drug, this represents a suitable tool to study the local effects of drugs on selected brain structures. Local administration of AMPA decreased the ACh output, but this effect was reversed by simultaneous perfusion with the GABA antagonist bicuculline. Moreover, the non-NMDA antagonist quisqualate perfused through the striatum, resulting in a decrease in the ACh output and a concomitant increase in the GABA output, suggesting that presynaptic AMPARs decrease the ACh output via a GABAergic intermediate [50].

The accumulated evidence on the role of presynaptic AMPARs in modulating neurotransmitter release from the axon terminals of various fiber types are summarized in Table 1.

### 3.2. AMPAR as Regulator of Axonal Development

During the early stages of neuronal development and synaptogenesis, a high number of long, thin and motile protrusions called filopodia are formed on both axons and dendritic branches of excitatory neurons. These protrusions are highly motile structures with short lifetimes, in the order of minutes to hours, and their motility is driven by actin filaments whose dynamic formation and regulation are modulated by ensembles of actin-binding proteins [56,57,58,59]. Axonal filopodia are the starting points of presynaptic specializations. In dendrites, some of these filopodia evolve into mushroom shape dendritic spines, which are the dynamically stable postsynaptic connection sites [60].

Several studies investigated whether AMPARs are involved in the regulation of dendritic dynamics. These studies showed that post-synaptic AMPAR activation by synaptically released glutamate is necessary for the maintenance of dendritic spines [61] and that the mechanism through which AMPAR activation changes dendritic spine density, shape and morphology is the regulation of actin dynamics [62,63,64].

While actin dynamics in dendritic filopodia have been extensively studied, their dynamics in axonal filopodia have been less investigated. Chang and De Camilli studied the role of axonal filopodia in the plasticity of the presynaptic compartment during synaptogenesis. They transfected in vitro hippocampal neurons with GFP-actin and observed that the activation of AMPARs/KARs inhibited filopodia motility and actin dynamics. The inhibitory effect on axon motility was mediated by activation of axonal glutamate receptors [65]. These results strongly suggest that glutamate controls plastic structural changes not only postsynaptically but also presynaptically.

### 3.3. Signaling Pathways of Presynaptic AMPAR

The molecular mechanism by which AMPARs regulate presynaptic function was partially clarified. Their classical function as ion channels may partially account for this effect. However, several studies highlighted that AMPARs also possess metabotropic properties. The first evidence concerning metabotropic AMPARs came from studies on postsynaptic AMPARs. These studies highlighted that postsynaptic AMPAR signaling in many neuron types involves a G-protein coupled to a protein kinase cascade that is independent from the ion channel activity [66,67,68,69,70,71]. To date, little is known about the downstream signaling involved in the metabotropic properties of presynaptic AMPARs. It has been reported that AMPAR stimulation in hippocampal neurons increases the levels of phosphorylated mitogen-associated protein kinase (MAPK) in all the neuronal compartments, including the axonal growth cones, and that this AMPA-mediated activation of MAPK occurs independently from ion flux through the receptor channel [72] (Figure 3). The direct involvement of AMPARs in MAPK activation was tested by silencing the expression of the GluA1,2,3 subunits with siRNA. The silencing of GluA2 and GluA3, but not of GluA1, abolished AMPA-induced MAPK activation, suggesting that these subunits are crucial for the activation of presynaptic MAPK. Furthermore, both in axonal growth cones and at mature presynaptic terminals, the activation of AMPARs led to MAPK-dependent phosphorylation of synapsin I at sites four and five, thus reducing the synapsin I-actin interaction [73] and causing the dispersion of synapsin I immunoreactivity in AMPA-treated cultured neurons. In parallel, AMPA also increased the rate of SVs recycling through a pathway involving MAPK activation [72]. The activation of metabotropic signaling cascades involving MAPK after presynaptic AMPAR stimulation [72] seems, therefore, to modulate the dynamics of vesicle fusion, thus regulating presynaptic plasticity.

MAPKs are important signal transducing enzymes that convert extracellular stimuli into a wide range of cellular responses; their activation by phosphorylation is achieved through a signaling cascade of three serially linked protein kinases [74]. The SV-associated protein synapsin I contributes to the modulation of SV exocytosis, by linking vesicles to each other and to the cytoskeleton, maintaining a reserve pool of SVs in the proximity of the release sites [75]. Synapsin I is phosphorylated at multiple sites by several kinases and is the major presynaptic substrate for MAPK: three MAPK-dependent phosphorylation sites are present in synapsyn I (P-sites 4/5 and P-site 6) and the phosphorylation of these sites regulates the interaction of the protein with the actin cytoskeleton [73,76,77]. The brain-derived neurotrophic factor (BDNF) leads to the activation of MAPK, leading to an increase in synapsin I phosphorylation, a release of the SVs from the cytoskeleton and a consequent increase in neurotransmitter release [77,78,79]. These events occur in both developing neurons and mature synapses; however, the stimulatory effect of AMPA seems to be prominent in immature neurons, as the number of presynaptic AMPARs decreases as neurons undergo maturation [72].

Another consequence of presynaptic AMPAR stimulation could be the modulation of a cAMP-dependent PKA signaling cascade. PKA is a key modulator of Ca^2+^-triggered vesicle fusion and several of its substrates are proteins involved in the modulation of synaptic function [80]. The phosphorylation of synapsin I at the PKA-dependent P-site 1 modulates its binding to SVs and regulates the efficiency of neurotransmitter release by increasing the availability of vesicles for recycling [79]. In addition to synapsin I, PKA modulates the function of the SV cysteine string protein (CSP), which is involved in the late stages of exocytosis by dynamically interacting with the two major exocytotic proteins, synaptotagmin I and syntaxin. CSP phosphorylation by PKA drastically decreases its affinity for these two proteins and slows SVs’ release while increasing the quantal size [81]. Another potential PKA target is Snapin: the PKA-phosphorylation of Snapin at serine 50, both in vitro and in vivo, increases its binding to synaptosomal-associated protein-25 (SNAP-25) and, consequently, enhances the association of synaptotagmin with the soluble N-ethylmaleimide-sensitive factor attachment protein receptor (SNARE) complex [82]. Hence, several lines of evidence show that presynaptic AMPARs play a role in the regulation of presynaptic plasticity (Figure 3). Further studies are required to better clarify the underlying molecular mechanisms.

### 3.4. Presynaptic AMPAR as Regulator of Synaptic Plasticity

Long-term synaptic plasticity is a crucial mechanism for learning and memory. The best studied forms of long-term synaptic plasticity consist in long-lasting modifications of synaptic strength that follow a prolonged or high frequency stimulation and can be either an enhancement (LTP) or a depression (LTD, long-term depression) of signal transmission. Both post- and pre-synaptic mechanisms contribute to the development of LTP or LTD. Indeed, changes in synaptic efficacy can occur at either side of the synapse, with distinct features: postsynaptic plasticity generally involves changes in postsynaptic receptor numbers or properties, whereas presynaptic plasticity involves an increase or decrease in neurotransmitter release [83,84].

Evidence shows that a change in the AMPAR-mediated transmission underlies several developmental and adult forms of synaptic plasticity. Early observations that glutamate increases its own release from the crayfish neuromuscular junction after the depolarization of the synaptic terminal [85] supported the idea that an increase in the firing rate increases glutamate release, which, in turn, increases the activation of presynaptic autoreceptors to locally modify terminal membrane conductance and/or polarization and thereby modulate excitability [86]. Similar plasticity phenomena have been reported to occur also in the CNS [87]. Thus, a positive feedback in which impulse activity enhances impulse related release has been described in several brain regions. As an example, long-term changes in the terminal excitability induced by the brief high frequency stimulation of the corticostriatal pathway have been described [88].

A wealth of reports have elucidated that in the post-synaptic compartment, whether an increase or a decrease in excitability is produced depends on the level and duration of membrane depolarization induced by the high-frequency stimulation. Similar considerations appear to apply to presynaptic mechanisms of plasticity. The level of membrane polarization, the relative activation of AMPARs/KARs, NMDARs and mGluRs subtypes and the subsequent increase in the intracellular levels of Ca^2+^ may result in a differential activation of Ca^2+^-activated enzymes, determining whether LTP or LTD is induced. Garcia-Munoz et al., using cortical tetanic stimulation followed by AMPA agonists and antagonists’ administration, showed that after the activation of AMPARs/KARs and mGluRs, the influx of Ca^2+^ through voltage-dependent channels and the subsequent activation of second messenger systems appear to play important roles in the induction and maintenance of LTP and LTD in cerebellum [89]. Moreover, AMPA-mediated MAPK phosphorylation pathways were found having a major role before and after synaptogenesis [17,73]. Thus, it is reasonable to expect that AMPAR metabotropic functions contribute to plasticity phenomena at the mature synapse. Further analysis will be required to validate these speculations.

## 4. Pathological Roles of Presynaptic AMPAR

### 4.1. Involvement of Presynaptic AMPAR in Axonal Pathology

Glutamate is responsible for the neuron death pathway named excitotoxicity, a mechanism that has been linked to several acute and chronic neurological conditions, such as stroke; traumatic brain injury; Alzheimer’s, Parkinson’s and Huntington’s diseases and Amyotrophic Lateral Sclerosis [90,91,92]. Evidence suggests that glutamatergic signaling is also directly involved in irreversible axonal injury. In particular, Ouardouz and coauthors demonstrated that functional glutamate receptors are expressed on myelinated dorsal column axons. They recorded intra-axonal [Ca^2+^] in myelinated rat dorsal column fibers using Oregon Green-488 BAPTA-1 fluorescence and found that both AMPAR and KAR agonists induced an increase in axoplasmic Ca^2+^-dependent fluorescence. Since ryanodine, a selective antagonist of ryanodine receptors presents on endoplasmic reticulum, significantly reduced the AMPAR-mediated axonal [Ca^2+^] rise, the authors suggested that AMPARs in dorsal column fibers operated via Ca^2+^-induced Ca^2+^ release, which is known to be dependent on ryanodine receptors. Immunohistochemistry confirmed the presence of AMPAR and KAR subunits clustered at the surface of myelinated axons at the internodal region. The study concluded that central myelinated axons express functional AMPARs and KARs and that these glutamate receptor-dependent signaling pathways may promote an increase in intra-axonal Ca^2+^ levels, potentially contributing to axonal degeneration [21].

### 4.2. Presynaptic AMPAR in Pain Transmission

iGluRs are localized on the cell bodies and on the peripheral processes of nociceptors. Moreover, nociceptive PAFs release glutamate, activating postsynaptic glutamate receptors on spinal cord dorsal horn neurons [93]. A recent paper has reviewed the current knowledge of the contribution of spinal AMPARs to the cellular mechanisms relating to chronic pain [94]. The evidence shows that glutamate receptors, including AMPARs, are expressed on PAF presynaptic terminals, where they regulate neurotransmitter release; in particular, the activation of AMPARs causes a decrease in glutamate release during action potential evoked synaptic transmission [20,33]. The presence of functional iGluRs on PAF terminals supports the hypothesis that these receptors can actively control and modulate the transmission of nociceptive information at the first synapse. AMPARs are also expressed in trigeminal ganglion cells; a recent study investigated the possibility that adenosine triphosphate (ATP) plays a permissive role in the activation of presynaptic AMPARs, thus inducing glutamate release from nerve terminals isolated from the rat trigeminal caudal nucleus (TCN) [95]. The authors isolated nerve endings from the rat TCN, loaded them with [^3^H]D-aspartate and measured radioactivity to assess [^3^H]D-aspartate release under different experimental conditions. They found that the spontaneous release of [^3^H]D-aspartate was stimulated by ATP and by the ATP analogue benzoylbenzoyl-ATP, the stimulation was prevented by the selective purinergic P2X7 receptor antagonist A438079. When synaptosomes were exposed to AMPA plus a purinoceptor agonist, the release exceeded that observed with ATP or benzoylbenzoyl-ATP alone. The selective AMPAR antagonist 2,3-dioxo-6-nitro-7-sulfamoyl-benzo[f]quinoxaline (NBQX) blocked this “excess” release. The authors concluded that the purinoceptor P2X7 expressed on glutamatergic nerve terminals in the rat TCN can mediate glutamate release directly and indirectly by facilitating the activation of presynaptic AMPARs. They also suggested that, through this mechanism, purine receptors can play roles in peripheral and central sensitization and are associated with migraine headache. Indeed, the high level of glial ATP that occurs during chronic pain states can promote the widespread release of glutamate as well as can increase the function of AMPARs. In this manner, ATP can contribute to the AMPAR activation involved in the onset and maintenance of the central sensitization associated with chronic pain [95].

### 4.3. Presynaptic AMPAR in the Auditory System

Excitatory synapses along the auditory pathway, from the cochlear hair cells to the auditory cortex, utilize L-glutamate as their primary neurotransmitter and postsynaptic NMDARs, AMPARs and KARs orchestrate the transfer of electrical signals generated in hair cells by acoustic waves. Furthermore, presynaptic iGluRs have been reported to exist in hair cells and the presynaptic terminals of auditory brainstem neurons [96]. Immunocytochemical analyses in the rat cochlea showed that the GluA4 subunit is expressed at the presynaptic membrane of hair cells [97]. Furthermore, functional AMPARs are also expressed in the calyceal nerve terminal (calyx of Held) of rodents (as reported in Section 2) [24]. Notably, in auditory neurons, the presynaptic AMPAR transmits the signal in a metabotropic manner [24,96]. Hence, pre- and postsynaptic AMPARs can be considered interesting targets for the development of novel therapeutic agents to treat hearing dysfunctions in a site- and cell-directed manner [96].

## 5. Conclusions

Presynaptic AMPARs belong to a class of presynaptic receptors that includes GABA_B_ receptors [98], endocannabinoid receptors [99], mGluRs [100], KARs [101], NMDARs [102] and many others [103]. Presynaptic ionotropic and metabotropic AMPARs can mediate feedback on synaptic neurotransmitter release in vivo through the different mechanisms summarized in Figure 4. One of the key mechanisms is the modulation of neurotransmitter release that can occur in diverse manners depending on the synapse types and developmental stages. Such changes in presynaptic activity represent important mechanisms for the space- and time-dependent selectivity of the modulation of information processing in the brain and may play an important role in a range of physiological and pathological processes.

The molecular mechanisms that mediate the positive or negative modulation of the release of the neurotransmitter at the various synapse types are complex and as yet not completely clarified. Thus, further studies are needed to identify the signaling cascades of AMPAR-induced presynaptic modulation. We could speculate that at least in certain instances, the differences are due to the ionotropic or metabotropic activity of the receptor.

Another interesting topic for further research is the potential role of the presynaptic AMPAR in axonal development, since early evidence suggests a key role of the presynaptic AMPAR in the maturation of growth cones. Recently developed technologies such as optogenetics or super-resolution light microscopy might provide tools that have been lacking in the past to explore these aspects.

Although the pharmacological profile of presynaptic AMPARs is as yet poorly defined, presynaptic AMPARs could be considered a promising pharmacological target. No neurological diseases have been found to be specifically associated with presynaptic AMPAR dysfunction. However, the most recent progresses in this field have shown that the modulation of the activity of AMPARs is effective in chronic pain therapy [94,95,104]. Worthy of notice is the recent literature showing that the enhancement of glutamate signaling by AMPAkines, a class of agents that specifically potentiate the function of AMPAR, reduce acute and chronic pain [104,105,106]. In addition, recent studies indicate that AMPAkines may have an antidepressant effect [107,108,109] and that they may influence memory in a mouse model of intellectual disability [110]. Although these studies did not distinguish between the activation of a presynaptic or postsynaptic AMPARs, it is likely that presynaptic AMPARs participate in the molecular mechanisms leading to the therapeutic benefit. 

Hence, drugs able to modulate a presynaptic AMPAR may be tested for improving cognition, learning and memory, and for dampening pathological phenomena such as axonal injury, hearing dysfunctions, depression and chronic pain.

## Figures and Tables

**Figure 1 cells-10-02260-f001:**
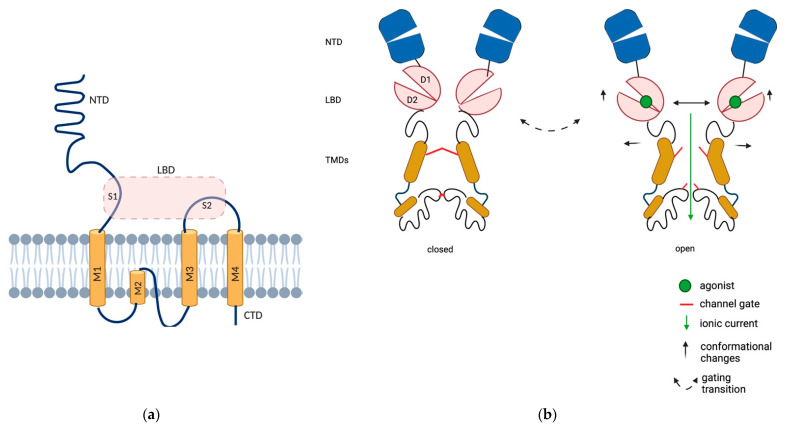
Structure and domain organization of AMPARs. (**a**) Representative structure of the AMPAR subunits. Each subunit contains an extracellular N-terminal domain (NTD, blue), a ligand binding domain (LBD, pink, which results from the proximity of two segments of amino acids, S1 and S2 domains), a transmembrane domain (TMD, orange) that forms the ion channel pore and a C-terminal cytoplasmic domain (CTD). The TMD contains three membrane-spanning helices (M1, M3 and M4) and a membrane re-entrant loop (M2). (**b**) Structural rearrangements in AMPARs during gating. The LBD exhibits a bilobate structure and captures the ligand in an interlobe cleft, as its D1 and D2 domains close similarly to a clamshell structure. LBDs of adjacent subunits dimerize back-to-back via their upper (D1) lobes, causing the separation of the lower D2 lobes upon ligand binding, which transmit mechanical force for the opening of the channel gate. Created in BioRender.com (accessed on 27 August 2021).

**Figure 2 cells-10-02260-f002:**
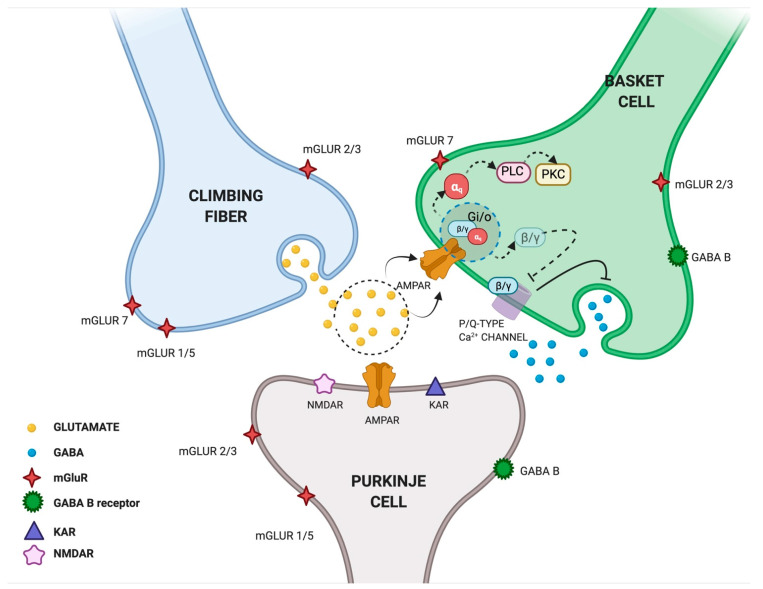
Glutamate spillover transmission between climbing fiber, Purkinje cells and Basket cells interneurons. Glutamate released from climbing fibers (CF) directly activates ionotropic AMPARs on Purkinje cells (PCs) eliciting postsynaptic excitation at CF-PC synapses. Glutamate released from CF-PC synapses then diffuses out of the synaptic cleft and acts on metabotropic AMPARs located in neighboring presynaptic terminals of interneuron Basket cells (BCs) resulting in the inhibition of GABAergic activity. In particular, AMPAR activation induces the dissociation of βγ subunits from G-proteins and inhibits the activity of P/Q-type Ca^2+^ channel in nerve terminals of cerebellar interneurons. Glutamate released from CF is shown in yellow, GABA released from BCs is shown in blue. Created in BioRender.com (accessed on 27 August 2021).

**Figure 3 cells-10-02260-f003:**
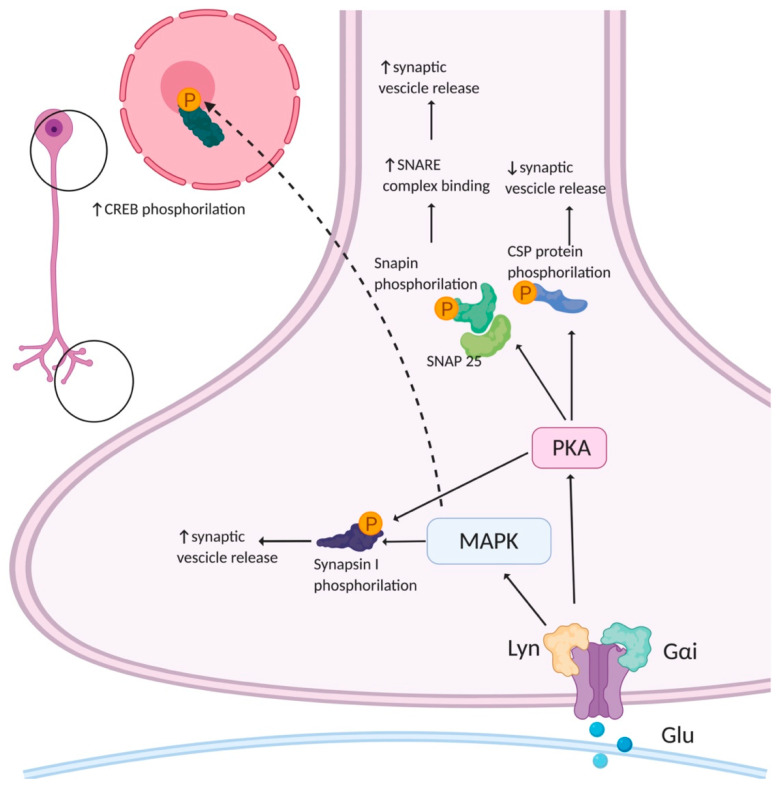
Potential presynaptic signaling cascades activated by AMPAR. AMPARs can activate mitogen-associated protein kinase (MAPK) and protein kinase A (PKA). MAPK phosphorylates synapsin I, an event that leads to an increase in evoked neurotransmitter release by increasing the availability of vesicles for fusion. PKA has many different substrates. Among them, Snapin phosphorylation enhances the association of synaptotagmin with the soluble N-ethylmaleimide-sensitive factor attachment protein receptor (SNARE) complex. PKA increases cysteine string protein (CSP) phosphorylation lowering its affinity for synaptotagmin I and syntaxin, slows vesicular release with a concomitant increase in quantal size. Created in BioRender.com (accessed on 27 August 2021).

**Figure 4 cells-10-02260-f004:**
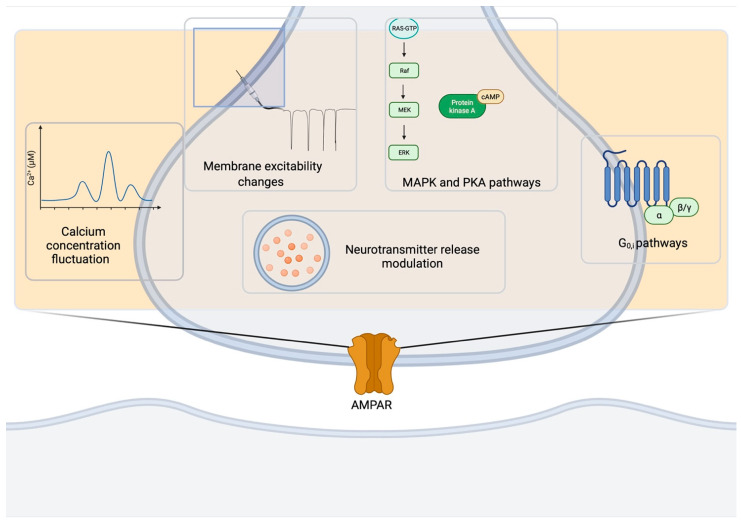
Basic mechanisms underlying AMPAR functions at the presynapse. AMPAR activity at the presynapse impinges on several pathways: fluctuations in [Ca^2+^], changes in membrane excitability, MAPK and PKA cascades, G_i/o_-activated pathways, neurotransmitter release modulation. Created in BioRender.com (accessed on 27 August 2021).

**Table 1 cells-10-02260-t001:** AMPAR as regulator of neurotransmitter release.

Neurotransmitter	Preparation	Methods	AMPAR Effect	Reference(s)
Glutamate	Rat forebrain slices	Loading of brain slices with [^3^H]D-aspartate and measurement of [^3^H]D-aspartate release in basal conditions and after administration of either L-glutamate, or racemic (R,S)-AMPA	L-glutamate and racemic (R,S)-AMPA dose-dependently enhanced [^3^H]-D-aspartate release	[31]
	Rat neostriatum, in vivo analysis	Intracerebral microdialysis	AMPA application increased Ca^2+^-dependent efflux of both [^3^H]-glutamate and [^14^C]-GABA in a dose-dependent manner	[32]
	PAFs of DRG neurons	Recording of receptor-mediated depolarization of the central terminals	Inhibition of glutamate release from primary afferent terminals in the superficial layers of the dorsal horn Induction of PAD	[33]
GABA	Cerebellar MLIs	Single-cell RT-PCR	Increase in GABAergic synaptic transmission	[19]
	Cerebellar interneurons	Measurement of GABA release after AMPAR activation by domoate	Potentiation of GABAergic synaptic activity in stellate cells	[37]
	Primary cultures of cerebellar neurons derived from GFP-GAD65 transgenic mice	Measurement of GABA release after AMPAR activation by AMPA and kainate Whole-cell recordings	Increase in the size of GABAergic terminals of interneurons Increase in spontaneous synaptic GABA release	[38]
	Presynaptic terminals of interneuron BC	Recording of IPSCs from PCs and measurement of intracellular Ca^2+^ fluctuations at BC terminals	Inhibition of GABAergic transmission through AMPAR-mediated activation of GTP-binding proteins (G_i/o_ proteins) and inhibition of the activity of P/Q-type Ca^2+^ channel	[39,40,41]
	Superficial dorsal horn of the rat spinal cord at second postnatal week Dorsal horn slices prepared from P27–P30 rat spinal cord	Recording of eIPSCs and mIPSPs	Increase in spontaneous GABA release Increase in mIPSP frequency Modulation of pain signaling	[42]
DA and NA	Purified synaptosomes from rat striatum	Superfusion with [^3^H]-tyrosine and measurement of newly synthesized [^3^H]DA release	Kainate and quisqualate facilitate DA release	[44,51,52]
	Hyppocampal synaptosomes	Superfusion system and a fluorimetric assay	AMPARs activation increase DA and NA release Increase in NA release in a Ca^2+^-dependent manner	[43]
	Rat hippocampus synaptosomes	Application of increasing AMPA concentrations	Release of preloaded [^3^H]NA in a Ca^2+^-dependent manner	[53]
	Hippocampal noradrenergic axon terminals	Activation of AMPAR	Induction of NA release	[28]
5HT	Rat brain cortex slices	Preincubation with [^3^H]5HT and superfusion in the presence of 6-nitroquipazine, Mg^2+^, KA or AMPA	Increase in 5HT release	[54]
	Hippocampal synaptosomes	Superfusion with AMPA	Ca^2+^-dependent DNQX-sensitive increase in 5HT release	[55]
ACh	Slices of rabbit caudate nucleus	Incubation with [^3^H]ACh and stimulation with NMDA, AMPA, L-glutamate or KA	Increase in ACh efflux	[47]
	Rat striatal slices	Incubation with [^3^H]ACh and treatment with 4-AP	Increase in ACh release	[48]
	Rat striatal slices and synaptosomes	Stimulation of NMDA and non-NMDA ionotropic receptors Application of AMPA	Facilitation of striatal Ach release Inhibition of ACh release	[49]
	Striatum of moving rats	Local administration of AMPA through a transversal microdialysis probe	Decrease in ACh output	[50]

## Data Availability

No new data were created or analyzed in this study. Data sharing is not applicable to this article.

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
