# Peer review of "Presynaptic AMPA Receptors in Health and Disease"

_cells, 2021, doi:10.3390/cells10092260_

Round 1

Reviewer 1 Report

This manuscript reviewed the accumulated evidence on the roles of presynaptic AMPAR in transmitter release, synaptic plasticity, and pain modulation. This work is a good summary for presynaptic AMPAR overall, but several caveats are also noticed.  

Major points: 

  1. The subtopics on the presynaptic AMPAR are loosely organized and partially overlapped. Consider revising the content, for instance, by reorganizing it into physiological functions (NT release, plasticity, NT spillover, axonal development, etc.), signaling pathways/molecular mechanisms,  and pathology (e.g., pain modulation, therapeutics, etc.).  
  2. The diverse mechanisms underlying the presynaptic AMPAR functions have been described in multiple places but not well-organized. They can be sorted into several categories at presynapses: Ca+ concentration changes, membrane excitability, MAPK or PKA pathway, Gi/o pathway, etc. A summary figure (s) to depict the significant mechanisms should also help readers. Many references/examples are mentioned for the release modulation of different transmitters, which may be summarized using a table. 
  3. It is vague about the most recent progress and unsolved questions critical on the presynaptic AMPAR.

Minor:

  1. Presynaptic AMPAR can increase or decrease NT release, depending on synapse types and development stages. Is any insight from the authors to explain this, and how does this impact therapeutics at system levels?
  2. New literature cited is limited: only 7 out of 88 references are from the recent ten years. Line 18 used “recent studies,” but it is unclear what that refers.    
  3. AMPARs are also expressed at the calyx of Held synapses in the auditory brainstem, where their activation induces inward currents, activates G-protein, and inhibits presynaptic Ca2+ currents (PMID: 15878995). The latter is essential because of the superliner relationship between transmitter release and Ca2+ across the entire physiological Ca2+ range (PMID: 15917809).
  4. Is any known disease relevant to presynaptic AMPAR dysfunction?   
  5. Fig-2 is mislabelled as fig-1. Further, the main point seems to attempt to show the Gi/o-mediated inhibition of GABA release from Basket-cell terminals containing AMPAR, but the issue was not evident in the figure. Many mGLURs draw at different places are distractive.
  6. The clarity of writing. For instance, the manuscript used the words like “regulate,” “control,” “influence” in multiple places, which reduced the clarity on whether it is positive or negative regulation. Line 387, the use of “Thus” is confusing.
  7. Line 349-350 statement on ” …mechanism… is mostly unknown” is not accurate. It contradicts the diverse mechanisms described in different synapses in the paper.

Author Response

attached file

Reviewer 2 Report

The review is mostly well-written and concise and should potentially be a useful addition to neuroscience field. However I do have some queries and suggestions.

  1. I think a short section on regulation of presynaptic AMPAR expression will be good to include. It can be combined with section 2 wherein the authors have mainly provided evidence for presence of presynaptic AMPARs. Thus, similar to lines 124-18 which somewhat indicates the developmental expression of presynaptic AMPARs, authors can include these references (Neuroscience. 2009 Jan 12;158(1):242-52 and Neuroscience. 2017 Mar 6;344:102-112.)
  2. Is there any evidence for involvement of presynaptic AMPARs in controlling glutamate spill-over and/or retrograde signalling. If so, authors should include references that have evaluated this.
  3. A small section on the physiology of presynaptic AMPAR in the auditory system (in addition to the nociception part; section 8) will be beneficial in accumulating the presynaptic AMPAR data in your single review as much as possible. There is a recent review that can be consulted. Hear Res. 2018 May;362:1-13.
  4. I think these key references should be included in the review according to their major findings.

Synapse-specific expression of functional presynaptic NMDA receptors in rat somatosensory cortex. Brasier DJ, Feldman DE. J Neurosci. 2008 Feb 27;28(9):2199-211.

Trafficking of presynaptic AMPA receptors mediating neurotransmitter release: neuronal selectivity and relationships with sensitivity to cyclothiazide. Pittaluga A, Feligioni M, Longordo F, Luccini E, Raiteri M. Neuropharmacology. 2006 Mar;50(3):286-96.

LTP of AMPA and NMDA receptor-mediated signals: evidence for presynaptic expression and extrasynaptic glutamate spill-over. Kullmann DM, Erdemli G, Asztély F. Neuron. 1996 Sep;17(3):461-74. doi: 10.1016/s0896-6273(00)80178-6.

Ultrastructural localisation and differential agonist-induced regulation of AMPA and kainate receptors present at the presynaptic active zone and postsynaptic density. Feligioni M, Holman D, Haglerod C, Davanger S, Henley JM. J Neurochem. 2006 Oct;99(2):549-60.

Calcium-permeable presynaptic AMPA receptors in cerebellar molecular layer interneurones. Rossi B, Maton G, Collin T. J Physiol. 2008 Nov 1;586(21):5129-45. doi: 10.1113/jphysiol.2008.159921.

Minor points:

  1. reference 61 and 69 are redundant
  2. please thoroughly check minor grammatical mistakes (e.g. line 6: Na, K, and Ca; line 59: an N-terminal; line 95: hippocampi, part of; combine lines 98-99 in the para above; line 155: Patel et al. were the first to suggest; line 177: with elevated; etc.)
  3. please provide the color codes for the domains in the legend of figure 1(a). Please correctly label the figures; numbering is confusing.
  4. please provide reference for lines 129-131

Author Response

attached file

Reviewer 3 Report

This review concerns the presynaptic AMPAR. Not much is known about this form of AMPAR. Thus, it is good to have a review manuscript that summarizes what have been published in presynaptic AMPAR.

I suggest that the authors could be more critical in the interpretation of the previous studies, and point out the future direction to reveal the location/trafficking/function of this special type of AMPAR.

Line 52-, it is stated that “AMPARs exist as homo- or heterotetramers of various combinations of the four subunits, thus giving rise to large receptor diversity”. While this statement is correct, the authors should mention in a few sentences that the GRIA1/2 and 2/3 are the most abundant forms in the brain. The authors may also mention their functional differences.

Line 56, “In particular, the presence of the GluA2 subunit can endow the channel with permeability to Ca2 “. Please includes the RNA editing of GRIA2 that alters Ca2+ permeability.

The authors may include a short paragraph to briefly describe the postsynaptic AMPAR, and the large number of associated proteins that regulate trafficking/localization/channel properties of the receptor. This is a prelude to the less known presynaptic AMPAR.

Line 47, “AMPAR are multimeric assemblies of GluA1-4 subunits”, further down on line 120 “Immunohistochemical characterization of GluR5 in dorsal columns revealed frequent punctate staining at the surfaces of neurofilament-positive axon cylinders”. GluR5 forms kainate receptor, but it is not introduced in the previous section. It may be better to define the protein names/gene names early on to avoid confusion.

Could the authors give some indication the spatial/brain region specific distribution of presynaptic AMPAR? and what is the abundance of the presynaptic AMPAR compared to classic postsynaptic AMPAR?

Many of the “presynaptic AMAPR” action may still be explained by the activity of the classic postsynaptic AMPAR. Could one exclude the possibility of retrograde transmission from PSD to presynapse? In the conclusion section, the authors should discuss the possible pitfall, and the probable improvement, of the previous studies. The future outlook of presynaptic AMPAR study should also be mentioned. It is not informative to state that “Although the pharmacological profile of presynaptic AMPAR is as yet poorly defined, presynaptic AMPARs could be considered a potential pharmacological target, and drugs able to modulate presynaptic AMPAR may be tested for improving cognition…”.

Author Response

attached file

Reviewer 4 Report

The review by Zanetti and colleagues on the function of AMPAR is a well-organized and well-written. The title however is somewhat misleading as there isn't sufficient discussion of AMPARs in disease and as such, I recommend revising the title rather than including more research on AMPARs in disease.

Minor comments:

  • There are some uses of abbreviations that have not been defined. Please carefully read the entire manuscript and define the first use of each abbreviation. Some include TM, M2, BDNF and P2X7.
  • Figure 1b - what does TM represent? Also, this schematic does not sufficiently illustrate the dimerization and and opening of the channel.
  • Figure 2 has been mislabelled as Figure 1
  • Figure 2 - the figure descriptor is wholly insufficient. The reader must be 'walked' through each of the components / different steps outlined in the figure.
  • Figure 2 - the 'representations' for AMPAR in purkinje and basket cells should be similar
  • Figure 2 - In the purkinje and basket cells, I believe these are NMDAR, AMPAR and KAR and not NMDA, AMPA and KAINATE?
  • Ln 269 - replace Ach with ACh
  • line 273 - change "[3H]DArelease" to [3H]DA release
  • There are other minor typographical errors that should be corrected.
  • Figure 3 has been mislabelled as Figure 2
  • Figure 3 is disproportionately large

Author Response

The review by Zanetti and colleagues on the function of AMPAR is a well-organized and well-written. The title however is somewhat misleading as there isn't sufficient discussion of AMPARs in disease and as such, I recommend revising the title rather than including more research on AMPARs in disease. We have modified the abstract, the Conclusions section and the main text in order to highlight the potential role of presynaptic AMPAR in disease and AMPAR as pharmacological target (lines 23-27 in the abstract, lines 637-706 in the main text, lines 730-747 in the conclusion).

Minor comments:

  • There are some uses of abbreviations that have not been defined. Please carefully read the entire manuscript and define the first use of each abbreviation. Some include TM, M2, BDNF and P2X7. We have defined the first use of each abbreviation, and we also introduced a list of abbreviations (pages 22-23).
  • Figure 1b - what does TM represent? we have modified the figure and legend; the transmembrane domains are now indicated as M1, M2, M3 and M4 (page 4).
  • Figure 2 has been mislabelled as Figure 1. We corrected the labelling of figure 2.
  • Figure 2 - the figure descriptor is wholly insufficient. The reader must be 'walked' through each of the components / different steps outlined in the figure. As requested, the have modified the legend of figure 2. The new legend text is as follows: Glutamate spillover transmission between climbing fiber, Purkinje cells and Basket cells interneurons. Glutamate released from climbing fibers (CF) directly activates ionotropic AMPARs on Purkinje cells (PC) eliciting postsynaptic excitation at CF-PC synapses. Glutamate released from CF-PC synapses then diffuses out of the synaptic cleft and acts on metabotropic AMPARs located in neighbouring presynaptic terminals of interneuron Basket cells (BC) resulting in the inhibition of GABAergic activity. In particular, AMPAR activation induces dissociation of βγ subunits from G-proteins and inhibits the activity of P/Q-type Ca2+ channel in nerve terminals of cerebellar interneurons. Glutamate released from CF is shown in yellow, GABA released from BC is shown in blue.
  • Figure 2 - the 'representations' for AMPAR in purkinje and basket cells should be similar. We corrected the figure as indicated.
  • Figure 2 - In the purkinje and basket cells, I believe these are NMDAR, AMPAR and KAR and not NMDA, AMPA and KAINATE? We corrected the figure as indicated.
  • Ln 269 - replace Ach with Ach. We modified the text as requested.
  • line 273 - change "[3H]DArelease" to [3H]DA release. We modified the text as requested.
  • There are other minor typographical errors that should be corrected. We modified the text as requested.
  • Figure 3 has been mislabelled as Figure 2. We corrected the labelling of figure 3.
  • Figure 3 is disproportionately large. We have reduced the size of figure 3.

Round 2

Reviewer 1 Report

recommended for publication.

Reviewer 3 Report

Could the authors remove "created in BioRender.com" from the figs? It is annoying.